# A Cross-Sectional Study of the Relationship Between Dietary Micronutrient Intake, Cognition and Academic Performance Among School-Aged Children in Taabo, Côte d’Ivoire

**DOI:** 10.3390/nu17223602

**Published:** 2025-11-18

**Authors:** Achil Tia, Amoin G. Konan, Jonas Hauser, Kouassi Y. Ndri, Olivier Ciclet, Lasme E. Esso, Charlemagne Nindjin

**Affiliations:** 1UFR Sciences et Technologies des Aliments, Université Nangui Abrogoua, Abidjan 02 BP 801, Côte d’Ivoire; charlemagne.nindjin@yahoo.fr; 2Centre Suisse de Recherches Scientifiques en Côte d’Ivoire, Abidjan 01 BP 1303, Côte d’Ivoire; georgette.konan@csrs.ci (A.G.K.); kouassiyvesrolandndri@gmail.com (K.Y.N.); docteuressoemmanuel@gmail.com (L.E.E.); 3UFR Biosciences, Université Félix-Houphouët-Boigny, Abidjan 22 BP 582, Côte d’Ivoire; 4NPTC Nutrition Konolfingen, Nestle Strasse 3, CH-3510 Konolfingen, Switzerland; jonas.hauser@rdls.nestle.com; 5Nestlé Institute of Food Safety and Analytical Sciences, Nestlé Research, Société des Produits Nestlé S.A., 1015 Lausanne, Switzerland; olivier.ciclet@rd.nestle.com

**Keywords:** micronutrient intake, dietary intake, cognition, academic performance, school-aged children, Côte d’Ivoire

## Abstract

Background: Adequate nutrient intake is crucial for cognitive development and academic performance in schoolchildren. This study assessed the association between dietary intake and both cognition and academic performance in school-aged children from Taabo, Côte d’Ivoire. Methods: A total of 252 schoolchildren aged 6–12 years were randomly selected from seven public primary schools. Dietary intake was assessed using repeated 24 h recalls. Cognitive skills were evaluated using Raven’s Colored Progressive Matrices (RCPM) and academic performance was determined based on end-of-year school results. Results: More than half of the participants had adequate intakes of vitamin A (55.8%), vitamin B6 (61.0%), vitamin B12 (61.0%) and omega-3 fatty acids (70.1%), while most had inadequate intakes of iron (96.8%), zinc (100%), thiamin (99.2%), riboflavin (99.6%) and folate (96.8%). Thiamin, riboflavin, folate, iron, zinc and a nutrient blend (comprising iron, zinc, thiamin, riboflavin and folate) showed significant associations with RCPM scores (*p* < 0.05). Additionally, higher intakes of thiamin (AOR = 6.3; 95% CI: 2.5–16.0, *p* < 0.001) and riboflavin (AOR = 2.2; 95% CI: 1.5–7.8, *p* = 0.003) were associated with increased odds of better cognitive performance compared with lower intakes. No consistent associations were found with academic performance. Conclusions: Compared to recommendations, intakes of several micronutrients were inadequate in most of the children. While thiamin, riboflavin, folate, iron, zinc and the nutrient blend showed significant associations with cognition, no association was found with academic performance. Further studies exploring such links are needed, especially those involving interventions.

## 1. Introduction

School age (6–12 years) represents a critical developmental period between childhood and adolescence, characterized by rapid cognitive growth due to social stimulation, including but not limited to the school environment [1]. Cognition, defined as the capacity to assimilate, process, and transform information into knowledge, underlies learning, memory and problem-solving skills that are essential for academic achievement [2]. Cognitive development is closely linked to brain structural maturation [3], which can be influenced by nutrition, highlighting the crucial role of nutrients in supporting cognitive performance [3].

Several nutrients play a crucial role in cognitive development. For instance, fatty acids, which constitute nearly 60% of the brain’s dry weight, serve as key structural components. A large body of evidence supports the beneficial effects of omega-3 fatty acids on the development of brain and mental functions [4,5]. Similarly, micronutrients such as iron, zinc, iodine, vitamin A, and B vitamins are essential for neurocognitive development. Studies have reported significant associations between these micronutrients and both cognitive function and academic performance in children [6,7,8,9].

Despite these established links, undernutrition remains a major public health concern in developing countries. In Côte d’Ivoire, national surveys primarily focus on children under five years of age; however, recent studies in school-aged children have reported high rates of malnutrition. The prevalence of undernutrition ranged from 19% to 40% among rural children aged 5–11 years [10], with 14.4% and 12.8% of children aged 7–15 years affected by moderate and severe chronic malnutrition, respectively [11]. In urban areas, 5.8% of schoolchildren were acutely malnourished and 26.7% chronically malnourished [12]. Other studies have reported anemia rates of 20–39%, iron deficiency in 47%, and vitamin A deficiency in 31% [12,13,14]. These findings demonstrate that multiple forms of malnutrition persist among Ivorian schoolchildren.

Furthermore, stunting has been shown to significantly influence academic performance [15,16], suggesting that the performance of these children may be associated with their dietary intake of the aforementioned nutrients or other factors not yet investigated. However, little is known, especially on the relationship between nutrient intake and cognitive function or academic performance in Ivorian schoolchildren, despite the high prevalence of malnutrition reported in these children. Therefore, this study was undertaken to assess the association between dietary intake and cognition as well as school performance in school-aged children from the Health and Demographic Surveillance System of Taabo, Côte d’Ivoire.

## 2. Materials and Methods

### 2.1. Study Design and Area

This study used a cross-sectional observational design. It was conducted between January and May 2023 in a sample of primary schoolchildren (6 to 12 years) from the Health and Demographic Surveillance System (HDSS) of Taabo, Côte d’Ivoire.

The HDSS is located in the south-central region of Côte d’Ivoire, approximately 150 km from Abidjan and 60 km from Yamoussoukro, the economic and political capitals of Côte d’Ivoire, respectively [17]. The HDSS covers an area of 980 km^2^ between 6°0′ and 6°20′ north latitude and 4°55′ and 5°15′ west longitude. The region has a tropical climate characterized by two rainy seasons (March to July and October to November) and two dry seasons (December to February and August to September). However, climate change has increasingly blurred the distinction between these seasons [18]. The average annual rainfall is about 284 mm, while the average temperature ranges between 27.6 °C and 28.3 °C. Relative humidity varies between 76.6% and 79.6%. The Taabo HDSS is predominantly rural and comprises 13 main villages, over 100 small hamlets, and a small town called Taabo-Citée with a population of 7514. The total population of the HDSS is estimated at 57,189. Agriculture is the dominant economic activity, with staple crops including yams, bananas, maize, and cassava, while cocoa and coffee are the main cash crops. Livestock (cattle, small ruminants, pigs, and poultry) and fishing also contribute to the local economy. This study was conducted in the small urban area (Taabo-cité) and in six other villages, which are Tokohiri, Ndenou, Ahouati, Leleblé, Ahouakro, and Adikouassikro.

### 2.2. Sample Size and Sampling Procedure

A total of 252 children aged 6 to 12 years were recruited in seven public primary schools (PPSs). The HDSS of Taabo is administratively divided into three academic zones: zone A, zone B, and zone C. Seven government-owned schools, namely Ahouakro, Tokohiri, N’Denou, Ahouati, Leleble, and Taabo-Barrage PPS, were randomly selected from these zones, with proportional consideration given to the number of schools in each zone. Then, 36 children from grades one to six were recruited in each school, with six children per grade. Participants were selected using a systematic random sampling method from school records provided by teachers or headteachers. The sample size (n) was determined based on the number of school-aged children in the selected schools, using the Yamane formula [2], as given below.(1)$$n=\frac{N}{\left(1+N\left(e^{2}\right)\right)}=\frac{681}{\left(1+681(0.05^{2}\right))}=251.99\;\approx \;252$$
where *N* is the total population of school-aged children in grades one to six (*N* = 681) and *e* is the margin of error, set at 0.05 (5%).

### 2.3. Inclusion and Exclusion Criteria

Children aged 6 to 12 years, who were enrolled in one of the seven participating schools, were eligible for inclusion. On the other hand, children were excluded if parental or guardian consent was not obtained, and if they were outside the age range, had any apparent disease or were enrolled in nutrition intervention studies.

### 2.4. Sociodemographics and Anthropometrics

Sociodemographic information collected included the age, sex, and school level of the participants. The level of education and the occupation of parents or guardians were also collected. The age of each child was determined based on their birth date provided in the school records.

For anthropometric data, the height and weight of each child were measured according to the recommendations of the World Health Organization for children and adolescents [19]. Z-scores for each of the nutritional indicators, including BMI for age, weight for age, and height for age, as well as the prevalence of wasting, stunting, and thinness, were calculated using the AnthroPlus software v1.0.4 available on the WHO website. All children below two standard deviations (SD) were considered underweight, stunted or wasted. Children between −2 SD and +2 SD were considered normal, while those above +2 SD were considered overweight, overgrown or obese.

### 2.5. Dietary Intake Assessment

Dietary intake data were collected on two weekdays. All foods and beverages consumed by participants at home (including breakfast, lunch, dinner, and snacks) and outside the home were collected. A pre-tested 24 h dietary recall questionnaire was used, following the methodology described by Gibson and Ferguson [20] in the WHO Dietary Assessment Manual. According to this method, two non-consecutive 24 h recalls were obtained from each child, with at least two days between interviews. The first interview involved all participants (*n* = 252), while 245 participants (97%) completed the second interview. Visual portion estimation tools, such as household measures and a photographic food atlas, were used during data collection to help participants recall portion sizes. Dietary intakes of iron, zinc, iodine, vitamin B6, folate, vitamin B12, vitamin A, eicosapentaenoic acid (EPA) and docosahexaenoic acid (DHA) were estimated for each child using NutriSurvey software 2007, which had been adapted to include local Ivorian foods [21]. The dietary patterns of the children included both animal and plant-based sources. Therefore, nutrient bioavailability for iron and zinc was assumed to be moderate, in line with WHO/FAO guidelines [22]. Nutrient adequacy was assessed using the Recommended Dietary Allowances (RDAs) provided by WHO/FAO in 2004 for micronutrients [22] and by the European Food Safety Authority for omega-3 fatty acids [23]. A nutrient blend comprising iron, zinc, thiamin, riboflavin and folate was constructed based on their established links to cognitive function among school-aged children in Sub-Saharan Africa [2,24]. Due to significant variability, nutrient intake was categorized into tertiles based on the distribution of each nutrient: lower (≤33.3rd percentile), medium (33.4th–66.7th percentile) and higher (>66.7th percentile).

### 2.6. Cognitive Skills Assessment

Cognitive skills were assessed using the Raven’s Colored Progressive Matrices, a widely used test of non-verbal intelligence. The RCPM was selected for its cultural fairness and ability to assess fluid intelligence, which is crucial for academic performance [2]. To ensure that all children understood the tasks, test instructions were first given in French and then translated into the local language. Individual test sessions were conducted without a time limit, following the standard procedure described in the test manual [25]. During the test, participants were presented with a matrix of symbols and asked to identify the missing symbol from six options. Each correct response was scored as one point, yielding a maximum possible score of 36. As no RCPM data exist for Ivorian children, cognitive performance was categorized as low (below the 50th percentile), medium (between the 50th and 75th percentiles) and high (at or above the 75th percentile) [26]. The corresponding threshold scores were as follows: low < 14, medium 14–18, and high ≥ 18.

### 2.7. Academic Performance Assessment

Academic performance was quantified using end-of-year results in mathematics and literature from the 2023–2024 school year. Both subjects were assessed in all grades (first through sixth). Evaluations were standardized in all participating schools, which minimized bias related to teacher grading. Performance was classified based on the school’s benchmark: low performance for scores below 5 out of 10, medium performance for scores between 5 and 7 out of 10, and high performance for scores above 7 out of 10 [27]. Noteworthy, in Côte d’Ivoire, the minimum required score for primary school is 5 out of 10.

### 2.8. Statistical Analysis

Data were analyzed using the Statistical Package for the Social Sciences version 20.0 (SPSS IBM Inc., Chicago, IL, USA). Categorical variables were summarized using frequencies and percentages, while continuous variables were expressed as means ± standard deviations (SD). Principal component analysis (PCA) was applied to identify nutrients associated with cognitive or school performance. As the outcome variables (cognition, mathematics and literature) were not normally distributed, performance was compared across sociodemographic and anthropometric groups using the Kruskal–Wallis test, with the Mann–Whitney U test applied for two-group comparisons. Chi-squared tests (χ^2^) were conducted to assess associations between categorical variables. A multivariable logistic regression was further conducted to identify factors associated with cognitive and school performance in the participants. Variables that were significantly (*p* < 0.05) associated with cognition or school performance in the bivariable analyses were selected for the regression models. All models were adjusted for maternal education level and father’s occupation, given that these factors were identified as significant confounders of the outcome variables. No correction for multiple comparisons was applied; instead, unadjusted *p*-values were considered, with significance set at *p* < 0.05.

## 3. Results

### 3.1. Sociodemographic and Anthropometric Characteristics

As shown in Table 1, 16.7% of children were enrolled in each school grade. Among the children recruited, 116 (46.2%) were female and 135 (53.8%) were male. The mean age was 9.3 ± 1.9 years (range: 6–12 years). Age distribution was as follows: 35.1% were aged 6–8 years, 30.3% were aged 9–10 years and 34.7% were aged 11–12 years. The majority of participants (80.5%) came from large households (more than five members), and most (97.2%) lived with their parents. Regarding parental characteristics, 79.3% of fathers worked in the primary sector (e.g., agriculture) and most had completed only primary education. In contrast, 60.6% of mothers were housewives with no formal education. Approximately 72.5% of the children reported eating lunch at the school cafeteria. Additionally, 90% revised their coursework for 30–60 min after school (classified as moderate study duration). More than half of the children (56.7%) had never repeated a school grade.

For anthropometrics, the mean z-scores were −0.49 (SD ± 1.15) for BMI-for-age, −0.32 (SD ± 1.19) for weight-for-age and −0.08 (SD ± 1.78) for height-for-age as compared with the WHO references (Appendix A). The BMI-for-age of the children was slightly lower (around 2% deviation) when compared with the WHO growth standards, whereas height-for-age and weight-for-age were substantially lower, with more than a 5% deviation (Appendix A). The prevalence of stunting, underweight and wasting was 11.6%, 10. 1% and 9.2%, respectively. The percentage of participants having any kind of undernutrition (stunting, wasting and underweight) was 30.9%. Boys were more affected by all forms of undernutrition than girls (Figure 1).

### 3.2. Cognitive and School Performance

The overall score obtained by the children on the cognitive test was 15.37 ± 4.38 (out of a maximum of 36), with the majority (43.4%) having low performance (score below the 50th percentile). For school performance, the overall scores were 5.96 ± 2.36 for mathematics and 5.43 ± 1.79 for literature (out of a maximum of 10 for both). The majority of the children in mathematics (44.2%) and literature (47.8%) had medium performance (Score between 5 and 7 out of 10) as shown in Appendix A.

### 3.3. Cognitive and Academic Performance by Sociodemographic Background

Table 1 presents cognitive and school performance across the different sociodemographic groups. A significant difference in cognitive performance was observed between boys and girls (*p* = 0.046), with boys performing better overall (15.8 ± 4.3) than girls (14.9 ± 4.5). The Kruskal–Wallis test also revealed significant differences across age groups (*p* < 0.001) and school grades (*p* < 0.001). After adjusting for covariates such as parental education and occupation, girls were less likely than boys to achieve high cognitive performance (AOR = 0.6; 95% CI: 0.3–0.9; *p* = 0.048). Moreover, cognitive performance improved with age: older children (11–12 years) were more than twice as likely to score above the 50th percentile compared with younger children (6–8 years and 9–10 years). Similarly, cognitive performance increased with grade level, as children in lower grades (Grades 1–4) had lower odds of achieving higher cognitive performance compared with those in Grade 6.

Regarding academic performance, significant differences were also observed between age groups and grades. In contrast to the trends noted in cognitive performance, younger children (6–8 years) achieved higher scores in both mathematics (*p* < 0.001) and literature (*p* = 0.002) compared with children aged 9–10 and 11–12 years. Similarly, children in lower grades (Grades 1 and 2) outperformed those in the upper grades. The adjusted logistic regression analysis revealed that Grade 2 children (AOR = 4.1; 95% CI: 1.3–12.7; *p* = 0.014) had approximately fourfold higher odds of achieving better performance in the literature compared with those in Grade 6.

### 3.4. Cognitive and Academic Performance by Anthropometric Characteristics

The results of cognitive and academic performance by anthropometric characteristics are presented in Appendix A. A significant difference was observed in mathematics performance across weight-for-age groups (underweight, normal weight and overweight; *p* = 0.035). However, in the regression analysis, no associations were found between weight-for-age groups and school or cognitive outcomes.

### 3.5. Associations Between Nutrient Intake and Cognition or Academic Performance

Dietary intake outcomes are summarized in Table 2. The majority of the children had inadequate intakes of vitamin B1 (99.2%), vitamin B2 (99.6%), folate (96.8%) and iron (96.8%). Zinc intake was inadequate in all the children (100%). However, vitamin A (55.8%), vitamin B12 (61.0%) and omega-3 fatty acids (70.1%) intakes were adequate in most of the children and iodine (100%) in all the participants.

The PCA showed that all assessed nutrients were found along one component with cognitive test scores, but not along the other component, which contained the two outcomes of school performance (Appendix A). In addition, chi-squared tests (χ^2^) revealed significant associations between thiamin (χ^2^ = 20.8; *p* < 0.001), riboflavin (χ^2^ = 12.6; *p* = 0.002), folate (χ^2^ = 10.4; *p* = 0.006), iron (χ^2^ = 7.1; *p* = 0.029), zinc (χ^2^ = 6.8; *p* = 0.033), nutrient blend (χ^2^ = 7.5; *p* = 0.024) intakes and cognitive test scores (Table 3). In multivariable regression analysis, higher tertiles of thiamin (AOR = 6.3; 95% CI: 2.5–16.0; *p* < 0.001) and riboflavin (AOR = 2.2; 95% CI: 1.5–7.8; *p* = 0.003) were associated with greater odds of better cognitive performance compared with lower tertiles.

Regarding academic performance, thiamin (χ^2^ = 7.9; *p* = 0.019) and zinc (χ^2^ = 9.7; *p* = 0.008) intakes were significantly associated with literature. Most children with lower intakes of thiamin (52.0%) and zinc (60.0%) had medium levels of performance, outperforming those with higher intakes. The adjusted analysis revealed inverse associations between both micronutrients and literature. Higher intakes of thiamin (AOR = 0.30; 95% CI: 0.10–0.60; *p* = 0.002) and zinc (AOR = 0.40; 95% CI: 0.20–0.70; *p* = 0.005) were associated with lower odds of achieving better literature performance (Table 4).

## 4. Discussion

This study reported the association of dietary intake with both cognition and school performance in school-aged children from the HDSS of Taboo, Côte d’Ivoire. Almost half (46.2%) of the participants were female, indicating that the sample of children recruited in the study was balanced in terms of gender. The results showed that cognitive performance increased with both age and school grade. This trend may reflect the combined effects of developmental maturation and sustained educational exposure. These findings are consistent with those of previous studies in school-aged children (4–7 years), which showed that age-related gains in cognitive abilities emerge progressively throughout childhood, facilitated by enriched learning environments and structured curricula [29]. Conversely, an inverse trend was observed for academic performance, with younger and lower-grade children outperforming their older peers in upper grades. These results may reflect a relative grade or age effect as documented in previous research [30,31]. Evidence suggests that younger children often benefit from greater developmental responsiveness, more tailored instructional support and simpler curricular demands in early grades [32]

Analysis of nutritional indicators showed that participants had lower BMI-for-age (Δ = −0.49) and weight-for-age (Δ = −0.32) compared with WHO standards. The overall prevalence of undernutrition was relatively high among the children in this study. Co-occurrence of stunting and underweight was present in 21.7%, contributing to an overall undernutrition prevalence of 30.9%. Similar findings have been reported in previous studies conducted among school-aged children in Côte d’Ivoire [10,14]. Child malnutrition remains a major challenge for many developing countries. According to the WHO report, 22.5% malnourished children live in Sub-Saharan Africa [33]. It is documented that malnutrition can negatively affect cognitive development in children, resulting in poor academic performance. A review conducted by Grantham-McGregor showed that malnourished children exhibit poorer cognitive abilities and academic achievement compared to well-nourished peers [34]. Another study in Kenyans primary school children reported a significant association between undernutrition and lower academic performance [35]. However, in the current study, although obese children performed better in mathematics than their underweight peers, no consistent associations were observed between nutritional status and either academic or cognitive outcomes. Further research may help to elucidate how nutritional status influences school performance in children from the HDSS of Taabo.

This study found a high percentage of inadequate intakes of thiamin, riboflavin, folate and iron, which suggests a widespread burden of micronutrient inadequacy among the children involved in this study. Nutrient intakes were principally assessed based on the Dietary Reference Intake provided by WHO/FAO [22]. Similar findings have been reported among children aged 7–11 years in Côte d’Ivoire, where the adequate intake of none of the assessed nutrients, including iron and zinc, was met. In that study, zinc intake reached 72% of the recommended daily intake (RDI), while iron intake covered only 31% to 45.4% [21]. Additional evidence from studies assessing biochemical markers in Ivorian school-aged children has shown high prevalence of anemia (30–50%), iron deficiency (47%) and vitamin A deficiency (17%) [12,13,14]. Such deficiencies are commonly associated with inadequate dietary intake, poor bioavailability from predominantly plant-based diets, and frequent exposure to parasitic infections (e.g., helminths and malaria). Micronutrient deficiencies are well-recognized contributors to impaired immune function and cognitive development in children. It has been estimated that approximately 200 million children in low- and middle-income countries failed to reach their full cognitive potential due to inadequate nutrition [35].

The prevalence of inadequate nutrient intake observed among the children involved in this study is worrying, and therefore, there is a need for nutritional strategies to improve their nutritional status. In contrast to most other micronutrients, where at least a significant proportion of children were deficient, all children had adequate iodine intake. This finding highlights the effectiveness of iodine fortification strategies (e.g., iodized salt) in ensuring sufficient dietary iodine intake in Côte d’Ivoire [36].

This study revealed that intakes of thiamin, riboflavin and folate were significantly associated with RCPM scores. Higher intakes of thiamin and riboflavin were associated with increased odds of better cognitive performance compared with lower intakes. These findings are consistent with results from a previous study involving the same group of school-aged children, in which folate biomarker was significantly correlated (*p* = 0.03) with cognitive scores [37]. Other studies conducted among Kenyan and Korean school-aged children reported similar associations between thiamin, riboflavin and folate intakes and improved cognitive performance [24,38]. However, this study found no associations between vitamin B6 or vitamin B12 and either cognition or school performance. Similar findings were observed for plasma biomarkers of both vitamins in our previous study, where pyridoxal 5′-phosphate (the active form of vitamin B6) and vitamin B12 concentrations were not significantly associated with cognitive performance (*p* < 0.05) [37]. The results of this study are also consistent with findings from Ghanaian schoolchildren (9–13 years), where no associations were observed between vitamin B6 and B12 intakes and RCPM scores [39]. However, studies conducted in Kenyan and Korean school-aged children reported significant associations between vitamin B6, vitamin B12 intakes and cognitive test scores [24,38]. This divergence of findings suggests an ambiguity surrounding the relationship between vitamins B6 and B12 and cognitive function, which further investigations might help to elucidate. B vitamins are involved in several biological processes. They serve as key coenzymes in the metabolism of carbohydrates, fats and proteins, facilitating energy production crucial for brain function and cognitive development [40], particularly during primary school years. The brain’s daily energy consumption is estimated at 20–25% of the total body energy expenditure [41]. B vitamins are also involved in DNA synthesis, gene expression regulation, and neurotransmitter synthesis [42]. Deficiencies in B vitamins can significantly affect cognitive development and school performance.

Significant associations were also observed between iron intake and cognitive test scores. Children with higher iron intake demonstrated better cognitive performance, with a greater proportion scoring in the upper percentile (above 50th) on the cognitive scale. This finding aligns with results obtained from iron biomarkers in the same participants, where transferrin (an Iron-binding protein considered the primary iron transport protein) showed a strong correlation with cognitive test scores (*p* = 0.0004) and ferritin (an Iron storage protein) demonstrated a similar trend (*p* = 0.06) [37]. This finding is also consistent with results from a randomized controlled trial in Kenyan school-aged children (mean age ~7.4 years), where higher iron intake was significantly associated with improved RCPM scores. At the biological level, iron is involved in several essential processes. It serves as a cofactor in numerous enzyme systems critical for energy metabolism and neurological function. Iron also plays a key role in neurotransmitter synthesis and the myelination of nerve fibers, both of which are vital for cognitive development [43]. Iron deficiency can considerably affect health status and cognitive functions.

This study found a positive association between zinc intake and cognitive performance but an inverse association with school performance. These findings highlight inconsistencies in the reported relationship between zinc and cognitive or academic outcomes, as noted in the previous literature [44]. Although evidence linking zinc status to cognitive function exists [39], most studies have reported negative or inconclusive associations. For example, a study among Kenyan school-aged children found a positive association between zinc intake and RCPM scores; however, a negative association was observed with RCPM scores in Ghanaian schoolchildren [2]. Further research is needed to clarify the nature of the association between zinc and both cognitive and academic performance.

This work revealed a significant association between the composite blend of micronutrients (vitamin A, thiamin, riboflavin, folate, zinc and iron) and cognitive performance, suggesting a potential synergistic effect on cognitive development in school-aged children. During school-age ages (6–12), executive functions (EF), including working memory, attention control, planning, and inhibition, undergo rapid refinement, primarily governed by prefrontal cortex (PFC) maturation. This neurodevelopmental trajectory relies on: Neurotransmitter modulation (e.g., dopamine, acetylcholine), synaptic pruning and plasticity, myelination and neuronal energy metabolism. Each nutrient in the blend contributes to EF development: Vitamin A regulates retinoic acid signaling critical for dopaminergic plasticity and synaptic function [44]. Thiamin facilitates glucose metabolism and supports neurotransmitter synthesis [45]. Riboflavin enhances mitochondrial energy production in high-demand PFC circuits [46]. Folate drives DNA methylation, neurogenesis, and monoamine synthesis [47]. Zinc modulates Brain-Derived Neurotrophic Factor (BDNF) expression and synaptic plasticity, and iron is essential for dopamine production, myelin formation and oxygen delivery [48]. Evidence from randomized controlled trials has shown positive effects of these micronutrients on EF in school-aged children. A study in Ethiopian schoolchildren demonstrated that high-dose vitamin A supplementation significantly improved working memory and the combination with iron showed enhanced benefit in vulnerable subgroups [49]. Noteworthy, the nutrients included in the blend were selected based on observed significant correlations in the dataset (except for vitamin A) and prior literature linking these nutrients to cognitive development. Future research should further explore causal mechanisms and validate this combination in controlled intervention settings.

In this study, no associations were observed between vitamin A, iodine or omega-3 fatty acids intakes and cognitive or academic outcomes. Similar results were reported in Ghanaian school-aged children (9–13 years), where vitamin A showed no association with cognition [40]. In addition, a study in South African schoolchildren (6–11 years) reported no association between omega-3 fatty acids intake and cognition [50]. In contrast, studies among school-aged children in Korea and Kenya school-aged children found significant correlations [24,39]. Other studies reported significant associations between omega-3 fatty acids or iodine intake and cognition in school-aged children [4,51]. These findings suggest that evidence on the association between vitamin A, iodine, and omega-3 fatty acid intakes and cognition remains inconclusive. Population-specific factors, including genetic variability and underlying nutritional deficiencies, may influence these relationships. A recent review on the importance of lipids for neurodevelopment in low and middle-income countries indicated that the metabolism of fatty acids in humans is affected by gene variants of Fatty Acid Desaturases (FADS), which are responsible for encoding the key enzymes (D5D and D6D) needed for fatty acid synthesis. The conversion of ALA into DHA and EPA and of LA into ARA is performed by FADS but with low efficiency. Specifically, only 8–20% of a dose of ingested ALA is converted into EPA and 0.5–9% into DHA [52] Further studies might elucidate the relationship between these nutrients and cognitive functions. Noteworthy, most of the nutrients assessed in the present study were associated with cognition but not with school performance, even though a previous study has reported links between improved cognition and better academic performance among sub-Saharan schoolchildren [53]. This inconsistency could be attributed to the multifactorial nature of academic performance, which is influenced by both cognitive and environmental factors.

This study has some limitations. Participants were interviewed directly about their dietary intake from the previous day using a 24 h dietary recall questionnaire. Visual portion estimation tools, such as commonly used household measures and a photographic food atlas, were used during data collection to help participants recall portion sizes and reduce reporting bias. However, the authors recognize that the 24 h recall method may not be the most reliable tool for this age group and that overestimation or underestimation of food portions is therefore possible. Additionally, the study did not assess total energy intake or consider the impact of dietary inhibitors, such as phytates, on iron absorption, both of which could have affected the interpretation of micronutrient adequacy. Furthermore, as this was a cross-sectional study, it cannot infer causality.

## 5. Conclusions

This study found that cognition-related micronutrients, particularly B vitamins and minerals, are largely inadequate among school-aged children in the Taabo HDSS, Côte d’Ivoire, except for vitamin A and iodine, which were assumed adequate in most participants. While intakes of thiamin, riboflavin, folate, iron, zinc and the nutrient blend were significantly associated with cognitive test scores, no consistent associations were observed between nutrient intakes and academic performance, in contrast to previous studies. Future studies using controlled trials are needed to evaluate the causal relationship between these nutrients and cognition or school performance in this population.

## Figures and Tables

**Figure 1 nutrients-17-03602-f001:**
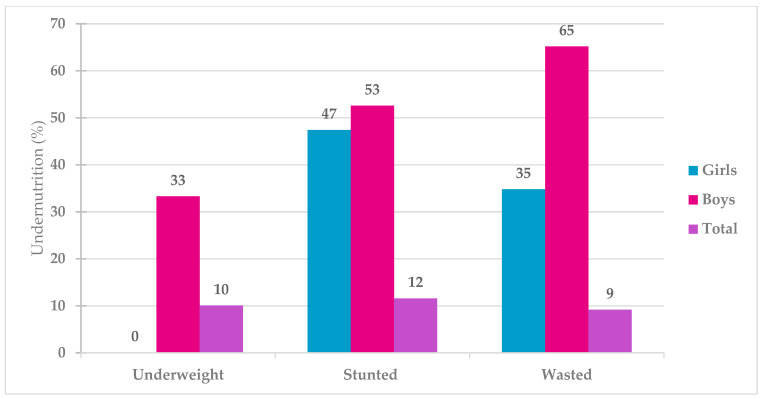
Prevalence of undernutrition among school-aged children in the Health and Demographic Surveillance System (HDSS) of Taabo, Côte d’Ivoire (*n* = 252).

**Table 1 nutrients-17-03602-t001:** Comparison of mean scores in cognitive and school performance across sociodemographic groups.

Sociodemographic Variables			Cognition		Mathematics		Literature	
	N	%	Mean ± SD	*p*-Value	Mean ± SD	*p*-Value	Mean ± SD	*p*-Value
**Gender**								
Female	116	46.2	14.9 ± 4.5 ^b^	0.046	5.9 ± 2.3	0.807	5.5 ± 1.8	0.624
Male	135	53.8	15.8 ± 4.3 ^a^		6.0 ± 2.4		5.4 ± 1.8	
**Age**								
6–8 years	88	35.1	13.1 ± 2.7 ^b^	<0.001	6.8 ± 2.8 ^a^	<0.001	5.9 ± 1.9 ^a^	0.002
9–10 years	76	30.3	15.9 ± 4.9 ^a^		5.4 ± 2.1 ^b^		5.4 ± 1.8 ^ab^	
11–12 years	87	34.7	17.1 ± 4.4 ^a^		5.6 ± 1.9 ^b^		5.0 ± 1.6 ^b^	
**Absence in class**								
Never	70	27.9	15.5 ± 4.9	0.807	6.0 ± 2.6	0.516	5.5 ± 2.0	0.874
1–3 days	157	62.5	15.3 ± 4.2		6.0 ± 2.2		5.4 ± 1.7	
More than 3 days	24	9.6	15.6 ± 3.8		5.4 ± 2.4		5.3 ± 1.8	
**Grade repetition**								
Never	141	56.2	15.3 ± 4.3	0.753	6.0 ± 2.5	0.114	5.7 ± 1.9 ^a^	0.04
Once	87	34.7	15.2 ± 3.9		6.1 ± 2.1		5.2 ± 1.6 ^a^	
More than once	23	9.2	16.5 ± 6.1		5.1 ± 2.2		4.9 ± 1.6 ^a^	
**Household size**								
Small (≤5)	49	19.5	15.7 ± 5.0	0.953	6.1 ± 2.3	0.682	5.6 ± 1.9	0.692
Large (>5)	202	80.	15.3 ± 4.2		5.9 ± 2.4		5.4 ± 1.8	
**Duration (min)**								
Short (≤30 min)	7	2.8	13.9 ± 2.3	0.284	6.6 ± 2.9	0.720	5.0 ± 1.6	0.284
Medium (30–60 min)	226	90.0	15.3 ± 4.3		5.9 ± 2.4		5.4 ± 1.8	
Long (≥60 min)	18	7.2	17.1 ± 5.4		6.2 ± 1.5		5.8 ± 1.6	
**School canteen**								
No	69	27.5	14.8 ± 4.4	0.085	5.8 ± 2.6	0.962	5.5 ± 2.1	0.966
Yes	182	72.5	15.6 ± 4.4		6.0 ± 2.3		5.4 ± 1.7	
**Live with parents**								
No	7	2.8	15.3 ± 4.6	0.930	4.8 ± 1.9	0.194	4.9 ± 1.1	0.534
Yes	244	97.2	15.4 ± 4.4		6.0 ± 2.4		5.4 ± 1.8	
**Mother’s occupation**								
Tertiary sector	24	9.6	15.9 ± 4.8	0.566	6.2 ± 2.3	0.680	5.4 ± 1.8	0.664
Housewife	227	90.4	15.3 ± 4.3		5.9 ± 2.4		5.4 ± 1.9	
**Mother’s education**								
Higher	5	2.0	17.8 ± 7. 1 ^a^	0.018	5.7 ± 1.2	0.419	5.8 ± 1.6	0.619
Illiterate	114	45.4	14.6 ± 4.2 ^a^		6.1 ± 2.4		5.4 ± 1.8	
Primary	88	35.1	15.8 ± 4.3 ^a^		5.7 ± 2.4		5.3 ± 1.8	
Secondary	44	17.5	16.3 ± 4.3 ^a^		6.1 ± 2.2		5.8 ± 1.9	
**Father’s occupation**								
Primary sector	200	79.7	15.2 ± 4.5	0.272	6.0 ± 2.4	0.322	5.4 ± 1.8 ^a^	0.023
Secondary sector	10	4.0	16.2 ± 5.2		4.9 ± 2.4		4.3 ± 1.2 ^b^	
Tertiary sector	41	16.3	15.8 ± 3.5		6.2 ± 2.3		5.9 ± 1.7 ^a^	
**Father’s education**								
Higher	21	8.4	16.2 ± 4.5	0.575	6.3 ± 2.3	0.688	5.9 ± 1.7	0.094
Illiterate	65	25.9	15.0 ± 4.5		6.1 ± 2.5		5.4 ± 1.7	
Primary	102	40.6	15.2 ± 4.4		5.8 ± 2.4		5.1 ± 1.8	
Secondary	63	25.1	15.8 ± 4.3		6.0 ± 2.3		5.8 ± 1.8	
**School grade**								
Grade 1	42	16.7	13.0 ± 2.0 ^b^	<0.001	6.8 ± 3.2 ^a^	<0.001	5.9 ± 2.0 ^a^	0.004
Grade 2	42	16.7	12.6 ± 2.4 ^b^		6.8 ± 2.4 ^ab^		5.9 ± 1.2 ^a^	
Grade 3	42	16.7	14.5 ± 3.3 ^b^		5.0 ± 2.1 ^c^		4.9 ± 1.9 ^b^	
Grade 4	41	16.3	16.1 ± 4.2 ^a^		5.2 ± 2.1 ^b^		5.0 ± 1.6 ^b^	
Grade 5	42	16.7	18.1 ± 5.5 ^a^		5.8 ± 1.7 ^ab^		5.3 ± 2.0 ^b^	
Grade 6	42	16.7	18.0 ± 4.4 ^a^		6.3 ± 1.8 ^ab^		5.5 ± 1.7 ^b^	

Values are presented as frequencies (*N*) and percentages (%) for sociodemographic variables, and as means ± standard deviations (SDs) for cognitive and school performance. Group comparisons were conducted using the Mann–Whitney U test (two groups) and the Kruskal–Wallis test (more than two groups). Values sharing the same letter within a column do not differ significantly according to Dunn’s post hoc test (*p* < 0.05).

**Table 2 nutrients-17-03602-t002:** Mean of daily nutrient intake, percentage of adequate and inadequate intakes, and distribution across tertiles.

Nutrient Intake	Mean ± SD	Adequate		Inadequate		1st Tertile		2nd Tertile		3rd Tertile	
		N	%	N	%	N	%	N	%	N	%
Vitamin A (µg)	1170.7 ± 1278.1	140	55.8	111	44.2	84	33.5	84	33.5	83	33.1
Vitamin B1 (mg)	0.4 ± 0.2	2	0.8	249	99.2	98	39.0	113	45	40	15. 9
Vitamin B2 (mg)	0.3 ± 0.2	1	0.4	250	99.6	94	37.5	112	44.6	45	17. 9
Folic acid (µg)	116.1 ± 67.3	1	0.4	250	99.6	84	33.5	85	33.9	82	32.7
Vitamin B12 (µg)	2.6 ± 1.8	153	61.0	98	39. 0	92	36.7	77	30.7	82	32.7
Vitamin B6 (mg)	3.6 ± 4.6	160	63.7	91	36.3	98	39.0	72	28.7	81	32.3
Iron (mg)	2.0 ± 1.7	8	3.2	242	96.8	83	33.1	85	33.9	83	33.1
Zinc (mg)	2.3 ± 1.0	0	0	251	100	85	33.3	86	34.3	80	31.9
Iodine (µg)	691.1 ± 416.9	251	100	0	0	83	33.1	85	33.9	83	33.1
Omega-3 fatty acids (g)	0.4 ± 0.4	176	70.1	75	29.9	128	51.0	61	24.3	62	24.7
Nutrient blend	123.0 ± 69.3	NA	NA	NA	NA	84	33.5	84	33.5	83	33.1

Note: Values are presented as mean ± standard deviation for daily nutrient intake, and as frequency (N) and percentage (%) for adequacy and tertile distribution. Nutrient adequacy was determined using WHO/FAO Dietary Reference Intakes [22], except for iron and omega-3 fatty acids, for which reference values from the Institute of Medicine [28] and the European Food Safety Authority [23] were used, respectively.

**Table 3 nutrients-17-03602-t003:** Distribution of cognitive and academic performance across nutrient intake groups.

Nutrient Intakes	Cognition			ꭓ^2^	*p*	Mathematics			ꭓ^2^	*p*	Literature			ꭓ^2^	*p*
	Above 75th	50th–75th	Below 50th			Above 7/10	5/10–7/10	Below 5/10			Above 7/10	5/10–7/10	Below 5/10		
**Vitamin A (µg)**															
Higher	27 (32.5)	30 (36.1)	26 (31.3)	3.4	0.181	18 (21.7)	40 (48.2)	25 (30.1)	3.6	0.166	14 (16.9)	33 (39.7)	36 (43.4)	3.0	0.228
Lower	22 (26.2)	24 (28.6)	38 (45.2)			29 (34.5)	36 (42.9)	19 (22.6)			15 (51.7)	43 (56.6)	26 (41.9)		
**Vitamin B1 (mg)**															
Higher	20 (50.0)	13 (32.5)	7 (17.5)	20.8	<0.001	8 (20.0)	20 (50.0)	12 (30.0)	2.5	0.274	5 (12.5)	13 (32.5)	22 (55.0)	7.9	0.019
Lower	18 (18.4)	24 (24.5)	56 (57.4)			33 (33.7)	42 (42.9)	23 (23.5)			18 (18.4)	51 (52.0)	29 (29.6)		
**Vitamin B2 (mg)**															
Higher	19 (42.2)	13 (28.9)	13 (28.9)	12.6	0.002	10 (22.2)	23 (51.1)	12 (26.7)	1.3	0.522	6 (13.3)	18 (40.0)	21 (46.7)	2.9	0.238
Lower	16 (17.0)	25 (26.6)	53 (56.4)			29 (30.9)	40 (42.6)	25 (26.6)			15 (16.0)	49 (52.1)	31 (31.9)		
**Folate (µg)**															
Higher	32 (39.0)	21 (25.6)	29 (35.4)	10.4	0.006	22 (26.8)	36 (43.9)	24 (29.3)	0.4	0.833	14 (17.1)	34 (41.5)	34 (41.5)	2.1	0.350
Lower	14 (16.7)	28(33.3)	42 (50.0)			26 (31.0)	34 (40.5)	24 (28.6)			13 (15.5)	44 (52.4)	27 (32.1)		
**Vitamin B12 (µg)**															
Higher	23 (28.0)	23 (28.0)	36 (43.9)	0.2	0.886	22 (26.8)	37 (45.1)	23 (28.0)	0.1	0.977	13 (15.9)	38 (46.3)	31 (37.8)	0.1	0.964
Lower	23 (25.0)	41 (44.6)	28 (30.4)			26 (28.3)	41 (44.6)	25 (28.3)			16 (17.4)	42 (45.7)	34 (37.0)		
**Vitamin B6 (mg)**															
Higher	18 (22.2)	21 (25.9)	42 (51.9)	1.5	0.482	18 (22.2)	41 (50.6)	22 (27.2)	3.2	0.207	12 (14.8)	38 (46.9)	31 (38.3)	0.7	0.719
Lower	25 (25.1)	31 (31.6)	42 (42. 9)			30 (30.6)	37 (37.8)	31 (31.6)			17 (17.4)	49 (50.0)	32 (32.6)		
**Iron (mg)**															
Higher	45 (30.0)	49 (32.7)	56 (37.3)	7.1	0.029	38 (25.3)	66 (44.0)	46 (30.7)	0.3	0.856	26 (17.3)	67 (42.1)	56 (37.3)	2.2	0.337
Lower	12 (14.5)	25 (30.1)	46 (55.4)			22 (26.5)	46 (47.0)	22 (26.5)			11 (13.3)	43 (51.8)	29 (34.9)		
**Zinc (mg)**															
Higher	30 (37.5)	25 (31.3)	25 (41.5)	6.8	0.033	20 (25.0)	38 (47.5)	22 (27.5)	1.2	0.548	12 (15.0)	29 (36.3)	39 (48.8)	9.7	0.008
Lower	17(20.0)	29 (34.1)	39 (45.9)			27 (31.8)	34 (40.0)	24 (28.2)			10 (11.8)	51 (60.0)	24 (28.2)		
**Iodine (µg)**															
Higher	26 (31.3)	24 (28.9)	33 (39.8)	1.0	0.605	20 (24.1)	38 (45.8)	25 (30.1)	2.0	0.366	10 (12.0)	45 (54.2)	28 (33.7)	2.9	0.236
Lower	21 (25.3)	29 (34.9)	33 (39.8)			28 (33.7)	35 (42.2)	20 (24.1)			17 (20.5)	36 (43.4)	30 (36.1)		
**Omega-3 fatty acids (g)**															
Higher	20 (32.3)	13 (21.0)	64 (46.8)	4.0	0.133	16 (25.8)	26 (41.9)	20 (32.3)	0.4	0.816	7 (11.3)	31 (50.0)	24 (38.7)	1.1	0.567
Lower	45 (23.8)	64 (33.9)	80 (42.3)			51 (27.0)	85 (44.9)	53 (28.0)			32 (16.9)	89 (47.1)	68 (36.0)		
**Nutrient Blend**															
Higher	31 (37.3)	22 (26.5)	30 (36.1)	10.3	0.006	23 (27.7)	36 (43.4)	24 (28.9)	0.3	0.849	15 (18.1)	34 (41.0)	34 (41.0)	2.2	0.329
Lower	13 (15.5)	29 (34.5)	42 (50.0)			26 (30.9)	33 (39.3)	25 (29.8)			13 (15.5)	44 (52.4)	27 (32.1)		

Note. Data are presented as frequency (*N*) and percentage (%). Associations of nutrient intakes with cognition and school performance were assessed using the chi-squared (χ^2^) test. Statistical significance was defined as *p* < 0.05. χ^2^ = chi-squared test statistic. *p =* chi-squared (χ^2^) test *p*-value.

**Table 4 nutrients-17-03602-t004:** Binary logistic regression of sociodemographic and nutritional factors associated with cognitive test scores and school performance above. the 50th percentile and 5 out of 10, respectively.

Variable	Cognition			Mathematics			Literature		
	AOR	95% CI	*p*-Value	AOR	95% CI	*p*-Value	AOR	95% CI	*p*-Value
**Gender**									
Girls	0.6	(0.3–0.9)	0.048	1.2	(0.7–2.1)	0.584	1.2	(0.7–2.0)	0.541
Boys	Reference								
**Age groups**									
6–8 years	0.4	(0.2–08)	0.010	1.9	(0.9–3.3)	0.073	1.8	(0.9–3.5)	0.103
11–12 years	2.2	(1.1–4.2)	0.024	1.1	(0.6–2.1)	0.784	0.5	(3.0–1.0)	0.054
9–10 years	Reference								
**School grade**									
Grade one	0.1	(0.5–0.4)	<0.001	0.8	(0.3–2.2)	0.59	1.2	(0.7–4.6)	0.235
Grade two	0.1	(0.–0.3)	<0.001	1.7	(0.5–5.8)	0.369	4.1	(1.3–12.7)	0.014
Grade three	0.2	(0.1–0.5)	<0.001	0.2	(0.1–0.6)	0.004	0.7	(0.3–1.6)	0.375
Grade four	0.4	(0.1–1.1)	0.072	0.3	(0.1–0.9)	0.029	0.4	(0.2–1.1)	0.064
Grade five	0.6	(0.2–1.6)	0.291	0.6	(0.2–1.6)	0.308	0.6	(0.3–1.5)	0.270
Grade six	Reference								
**Weight-for-age**									
Under/overweight	Reference								
Normal	0.4	(0.1–1.1)	0.066	1.7	(0.6–5.0)	0.342	1.7	(0.6–5.1)	0.332
**Vitamin B1 intake**									
Lower	Reference								
Higher	6.3	(2.5–16.0)	<0.001	0.7	(0.3–1.6)	0.676	0.3	(0.1–0.6)	0.02
**Vitamin B2 intake**									
Lower	Reference								
Higher	2.2	(1.5–7.8)	0.003	1.0	(0.4–2.2)	0.961	0.5	(0.3–1.1)	0.099
**Folate intake**									
Lower	Reference								
Higher	1.8	(0.9–3.3)	0.086	1.0	(0.5–2.0)	0.935	0.7	(0.3–1.2)	0.196
**Iron intake**									
Lower	Reference								
Higher	1.9	(1.0–3.6)	0.056	0.9	(0.4–1.8)	0.727	0.9	(0.5–1.7)	0.750
**Zinc intake**									
Lower	Reference								
Higher	1.7	(0.8–3.3)	0.130	1.1	(0.5–2.2)	0.820	0.4	(0.2–0.7)	0.005
**Nutrient blend intake**									
Lower	Reference								
Higher	1.7	(0.9–3.3)	0.101	1.0	(0.5–2.1)	0.918	0.6	(0.3–1.2)	0.136

AOR: Adjusted Odds Ratio; CI: Confidence Interval. Micronutrient intake levels were categorized into lower and higher tertiles, with the lower tertile used as reference. All models were adjusted for maternal education level and paternal occupation. Statistical significance was set at *p* < 0.05.

## Data Availability

The data presented in this study are available on request from the corresponding author due to ethical reasons.

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
