# Peer review of "A Cross-Sectional Study of the Relationship Between Dietary Micronutrient Intake, Cognition and Academic Performance Among School-Aged Children in Taabo, Côte d’Ivoire"

_nutrients, 2025, doi:10.3390/nu17223602_

Round 1
Reviewer 1 Report
Comments and Suggestions for Authors
This is a well-written and thoroughly conducted study in a novel population. However, I do have a few questions:
Sample size: The measurement of micronutrients by dietary recall is generally considered to be problematic, and perhaps magnified when conducting recalls in children. With that said, how did the authors arrive at a sample size of 252? How representative is this sample outside of the geographic area of study?
Statistical analyses: Why did the authors choose non-parametric tests, such as Mann-Whitney U and Kruskal-Wallis? What was the justification of adjustments for maternal education and paternal occupation?
Author Response
Comments 1: Sample size: The measurement of micronutrients by dietary recall is generally considered to be problematic, and perhaps magnified when conducting recalls in children. With that said, how did the authors arrive at a sample size of 252? How representative is this sample outside of the geographic area of study?
Response 1:
Thank you for pointing this out. We agree with this comment. Therefore, we have provided the following information: “The sample size was calculated based on the population of school-aged children in public primary schools involved in the study, using Yamane’s formula (Yamane, 1967), given as followed N= N/(1+N(e²)) =+681/(1+681(0.05²)) =251.99 ≈ 252. Where N is the total population of school-aged children in grades one to six (N = 681) and e is the margin of error, set at 0.05 (5%).” These details are provided in the revised manuscript at lines 107-112, Page 2
Comments 2: Statistical analyses: Why did the authors choose non-parametric tests, such as Mann-Whitney U and Kruskal-Wallis? What was the justification of adjustments for maternal education and paternal occupation?
Response 2: We have, accordingly, revised to emphasize this point as followed. “We used non-parametric tests because the outcome variables (cognitive test score, mathematics, and literature) were not normally distributed. The Shapiro-Wilk test revealed p-values less than 0.05 for all three variables, indicating a departure from normality. All models were adjusted for maternal education level and father’s occupation, given that these factors were identified as significant confounders of the outcome variables (cognitive test scores and/or school performance). Lines 194-198 and 204 to 207 – page 5.
Reviewer 2 Report
Comments and Suggestions for Authors
The study is timely and relevant, particularly given the burden of undernutrition in Sub-Saharan Africa. The use of standardized cognitive testing and dietary recall methods adds strength to the work. However, I believe that several aspects of the manuscript should be improved before it can be considered for publication.
1] While comprehensive, the introduction is overly long and contains repeated explanations of micronutrient functions. This could be shortened and focused more on the study gap in Côte d’Ivoire and the novelty of your research.
2] Please also ensure consistency in references (e.g., missing spaces in [9]have).
3] The choice of school grades in mathematics and literature as proxies for academic performance needs stronger justification, as grading systems can be context-specific. Please elaborate on why this is a valid measure.
4] “Tableau” is used instead of “Table” in multiple places.
5] The manuscript requires careful editing for English grammar and clarity. Examples include: “Iron is also key nutrients”, “Iodine is critical nutrient for metal functions”, “Iron supplementation was has also been associated…”.
Author Response
Comments 1: While comprehensive, the introduction is overly long and contains repeated explanations of micronutrient functions. This could be shortened and focused more on the study gap in Côte d’Ivoire and the novelty of your research.
Response 1: We agree with this comment. Therefore, we have shortened and focused the introduction more on the study gap in Côte d’Ivoire and the novelty of our research. Modifications were made from lines 43 to 66, page 2.
Comments 2: Please also ensure consistency in references (e.g., missing spaces in [9] have)
Response 2: Thank you for pointing this out. We have accordingly revised the manuscript.
Comment 3: The choice of school grades in mathematics and literature as proxies for academic performance needs stronger justification, as grading systems can be context specific. Please elaborate on why this is a valid measure.
Response 3: We totally agree with this comment. We have provided justification in the revised manuscript from line 181-183, Page 5 as followed” Academic performance was quantified based on mathematics and literature end-of-year results for the 2023–2024 school year. These two subjects were assessed in all grades (grade one to grade six). These evaluations were standardized in all schools participating in the study. This ensured consistency and minimized bias related to teacher grading.
Comment 4: “Tableau” is used instead of “Table” in multiple places.
Response 4: Thank you for pointing this point. We have carefully revised table titles accordingly.
Comment 5. The manuscript requires careful editing for English grammar and clarity. Examples include: “Iron is also key nutrients”, “Iodine is critical nutrient for metal functions”, “Iron supplementation had also been associated…”.
Response 5: We have carefully revised the manuscript accordingly. Thank you very much for pointing these out.
Round 2
Reviewer 2 Report
Comments and Suggestions for Authors
The authors have adequately responded to the comments.
Please check, some references are inconsistently formatted.
Author Response
Response to Reviewer Comments
- Reviewer 1
Comment 1: Does the introduction provide sufficient background and include all relevant references? Can be improved.
Response 1: We agree with this comment. The introduction has been revised to improve clarity, readability and background coverage. Please see the revised version in Lines 42-74, Page 1.
Comment 2: Are the methods adequately described? Can be improved.
Response 2: Thank you for this observation. The description of the methods has been improved. Please refer to the following revised sections:
- Sample size and sampling procedure (Lines 97-106)
- Inclusion and exclusion criteria (Lines 111-114)
- Dietary intake assessment (Lines 129-150)
- Cognitive skills assessment (Lines 152-154)
- Academic performance assessment (Lines 165-167)
(See Pages 2–3 of the revised manuscript.)
Comment 3: Are the results clearly presented? Can be improved.
Response 3: We fully agree with this comment. The presentation of the results has been improved for better readability. Please see the revised sections from Lines 209-266 (Pages 4-5). All modifications are highlighted in red color.
Comment 4: Are the conclusions supported by the results? Can be improved.
Response 4: The conclusion has been improved, particularly attention has been given to clarity (see Lines 460-467, Page 9).
Comment 5: Are all figures and tables clear and well-presented? Can be improved.
Response 5: Thank you for this suggestion. All tables and figures have been reformatted and improved for clarity.
Please see Pages 6-7, Page 10, and Pages 11-12 of the revised manuscript.
Noteworthy, the original Figure 1 (showing the study area) has been removed since it was published in a previous paper. A new Figure 1 has been added in the Results section (see Page 8).
Comment 6: Please check, some references are inconsistently formatted.
Response 6: All references have been carefully reviewed and reformatted.
- Quality of English Language
- Reviewer 2, Round 1
Comment: The English could be improved to more clearly express the research.
Response: We agree with this observation. The manuscript has undergone language editing to improve clarity and overall readability.
- Quality of Figures
Reviewer 2, Round 1
Comment: Figures and tables can be improved.
Response: All figures and tables have been reformatted and improved for better readability and presentation quality. Please see Pages 6-7, Page 10, and Pages 11-12 of the revised manuscript. The previous Figure 1 (study area) has been removed because it was recently published in another paper. A new Figure 1 has been added in the Results section (see Page 8).
